# Learning Non-Metric Visual Similarity for Image Retrieval

## Abstract

Measuring visual (dis)similarity between two or more instances within a data distribution is a fundamental task in many applications, especially in image retrieval. Theoretically, non-metric distances are able to generate a more complex and accurate similarity model than metric distances, provided that the non-linear data distribution is precisely captured by the similarity model. In this work, we analyze a simple approach for deep learning networks to be used as an approximation of non-metric similarity functions and we study how these models generalize across different image retrieval datasets.

## 1 Introduction

For humans, deciding whether two images are visually similar or not is, to some extent, a natural task. However, in computer vision, this is a challenging problem and algorithms do not always succeed in matching pictures that contain similar-looking elements. This is mainly because of the well-known *semantic gap* problem, which refers to the difference or gap between low-level image pixels and high-level semantic concepts. Estimating visual similarity is a fundamental task that seeks to break this semantic gap by accurately evaluating how alike two or more pictures are. Visual similarity is crucial for many computer vision areas including image retrieval, image classification and object recognition, among others.

Given a query image, content-based image retrieval systems rank pictures in a dataset according to how similar they are with respect to the input. This can be broken into two fundamental tasks: 1) computing meaningful image representations that capture the most salient visual information from pixels and 2) measuring accurate visual similarity between these image representations to rank images according to a similarity score.

In the last years, several methods to represent visual information from raw pixels in images have been proposed, first by designing handcrafted features such as SIFT Lowe (2004), then by compacting these local features into a single global image descriptor using different techniques such as Fisher Vectors Perronnin et al. (2010) and more recently by extracting deep image representations from neural networks (Babenko et al. (2014)). However, once two images are described by feature vectors, visual similarity is commonly measured by computing a standard metric between them. Although regular distance metrics, such as Euclidean distance or cosine similarity, are fast and easy to implement, they do not take into account the possible interdependency within the dataset, which means that even if a strong nonlinear data dependency is occurring in the visual collection, they might not be able to capture it. This suggests that learning a similarity estimation directly from visual data can improve the performance on image retrieval tasks, provided that the likely nonlinearity dependencies within the dataset are precisely learned by the similarity function.

Visual similarity learning is closely related to distance metric learning. Traditionally, distance metric learning algorithms were based on linear metrics such as the Mahalanobis distance. However, if the visual data presents any nonlinear interdependency, better results are expected when using nonlinear approaches. According to some studies Tan et al. (2006), standard metric axioms are not valid for human perception of visual similarity and hence, visual similarity functions should not necessarily satisfy distance metric conditions. Deep learning-based similarity learning methods are mostly focused on learning an optimal mapping from pixels to a linear space in which Euclidean distance can be applied. Instead, we propose a simple approach based on neural networks to learn a non-metric similarity score in the feature space.

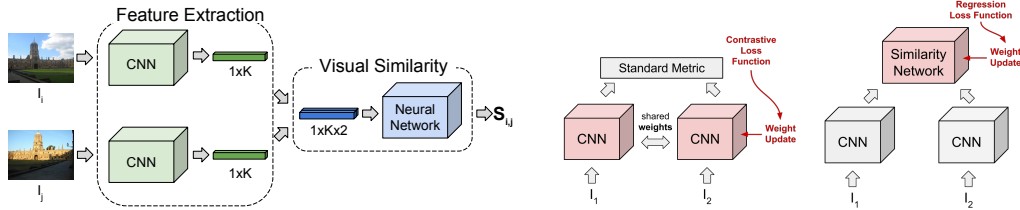

Figure 1: System overview. The feature extraction block computes visual representations of images whereas the visual similarity block estimates a similarity score using a neural network.

Figure 2: Siamese architectures (**left**) map pixels into high-quality vector representations. Our similarity network (**right**) learns a similarity function on top of the vector representations.

Figure 1 shows an overview of the proposed approach. By training a deep learning model, we can estimate a visual similarity function that outperforms methods based on standard metric computations. One convolutional neural network extracts image representations from input images, while a second neural network computes the visual similarity score. The visual similarity neural network is trained using both pairs of similar and dissimilar images in three stages. The output score of the similarity network can be directly applied as a similarity estimation to rank images in an image retrieval task. Experimental results on standard datasets show that our network is able to discriminate when a pair of images is similar or dissimilar and improve standard metrics score on top of that.

## 2 RELATED WORK

**Content-Based Image Retrieval.** Content-based image retrieval searches for images by considering their visual content. Given a query image, pictures in a collection are ranked according to their visual similarity with respect to the query. Early methods represent the visual content of images by a set of hand-crafted features, such as SIFT Lowe (2004). As a single image may contain hundreds of these features, aggregation techniques like bag-of-words (BOW) Sivic et al. (2003), Fisher Vectors Perronnin et al. (2010) or VLAD Jégou et al. (2010) encode local descriptors into a compact vector, thereby improving computational efficiency and scalability. Recently, because of the latest advancements on deep learning, features obtained from convolutional neural networks (CNN) have rapidly become the new state-of-the-art in image retrieval.

**Deep Learning for Image Retrieval.** Deep image retrieval extracts activations from CNNs as image representations. At first, some methods Babenko et al. (2014); Sharif Razavian et al. (2014); Wan et al. (2014); Liu et al. (2015) proposed to use representations from one of the last fully connected layers of networks pre-trained on the classification ImageNet dataset Russakovsky et al. (2015). When deeper networks such as GoogLeNet Szegedy et al. (2015) and VGG Simonyan & Zisserman (2014) appeared, some authors Babenko & Lempitsky (2015); Yue-Hei Ng et al. (2015); Sharif Razavian et al. (2014); Xie et al. (2015) showed that mid-layer representations obtained from the convolutional layers performed better in the retrieval task. Since then, there have been several attempts to aggregate these high-dimensional convolutional representations into a compact vector. For example, Gong et al. (2014); Yue-Hei Ng et al. (2015) compacted deep features by using VLAD, Mohedano et al. (2016) encoded the neural codes into an histogram of words, Babenko & Lempitsky (2015); Kalantidis et al. (2016) applied sum-pooling to obtain a compact representation and Razavian et al. (2016); Tolias et al. (2016) aggregated deep features by max-pooling them into a new vector. A different approach is to train the network to directly learn compact binary codes end-to-end (Erin Liong et al., 2015; Lin et al., 2015). Some authors have shown that fine-tunning the networks with similar data to the target task increases the performance significantly (Babenko et al., 2014; Gordo et al., 2016; Radenović et al., 2016; Salvador et al., 2016; Gordo et al., 2017). Finally, recent work has shown that adding attention models to select meaningful features can be also beneficial for image retrieval (Jiménez et al., 2017; Noh et al., 2017).

All of these methods are focused on finding high quality features to represent visual content efficiently and visual similarity is computed by simply applying a standard metric distance. General metrics, such as Euclidean distance or cosine similarity, however, might be failing to consider the inner data structure of these visual representations. Learning a similarity function directly from data may help to capture the human perception of visual similarity in a better way.

**Similarity Learning.** Some of the most popular similarity learning work, such as OASIS Chechik et al. (2010) and MLR McFee & Lanckriet (2010), are based on linear metric learning by optimizing the weights of a linear transformation matrix. Although linear methods are easier to optimize and less prone to overfitting, nonlinear algorithms are expected to achieve higher accuracy modeling the possible nonlinearities of data. Nonlinear similarity learning based on deep learning has been recently applied to many different visual contexts. In low-level image matching, CNNs have been trained to match pairs of patches for stereo matching Zagoruyko & Komodakis (2015); Luo et al. (2016) and optical flow Fischer et al. (2015); Thewlis et al. (2016). In high-level image matching, deep learning techniques have been proposed to learn low-dimensional embedding spaces in face verification Chopra et al. (2005), retrieval Wu et al. (2013); Wang et al. (2014), classification Hoffer & Ailon (2015); Qian et al. (2015); Oh Song et al. (2016) and product search Bell & Bala (2015), either by using siamese Chopra et al. (2005) or triplet Wang et al. (2014) architectures.

In general, these methods rely on learning a mapping from image pixels to a low dimensional target space to compute the final similarity decision by using a standard metric. They are designed to find the best projection in which a linear distance can be successfully applied. Instead of projecting the visual data into some linear space, that may or may not exist, our approach seeks to learn the non-metric visual similarity score itself. Similarly, Li et al. (2014) and Han et al. (2015) used a CNN to decide whether or not two input images are a match, applied to pedestrian reindentification and patch matching, respectively. In these methods, the networks are trained as a binary classification problem (i.e. same or different pedestrian/patch), whereas in an image retrieval ranking problem, a regression score is required. Inspired by the results of Wan et al. (2014), which showed that combining deep features with similarity learning techniques can be very beneficial for the performance of image retrieval systems, we propose to train a deep learning algorithm to learn non-metric similarities for image retrieval and improve results in top of high quality image representation methods.

## 3 Learning Visual Similarity

### 3.1 Definition

Visual similarity is the task that measures how related two images are by using their visual content. Given $n$ samples in the training image collection $I$, for each image $I_i \in I$ with $i \in [1, n]$, a global $d$-dimensional representation $x_i \in \mathbb{R}^d$ is obtained as $x_i = f(I_i, w_f)$, where $f$ is the function that maps images into global features and $w_f$ is the set of parameters of $f$. We define $s_{i,j}$ as the similarity score which measures how alike two images $I_i$ and $I_j$ are. The higher $s_{i,j}$ is, the more similar $I_i$ and $I_j$ are. The aim is to learn a visual similarity function $S$ that computes the similarity score from global image representations as:

$$s_{i,j} = S(x_i, x_j) = g(f(I_i, w_f), f(I_j, w_f), w_g)$$
$$s.t. \quad s_{i,j} > s_{i,k} \rightarrow I_i, I_j \text{are more similar than } I_i, I_k \tag{1}$$

where $g$ is a nonlinear function and $w_g$ is the set of parameters to optimize.

Note that $g$ does not have to be a metric in order to be a similarity function and thus, it is not required to satisfy the rigid constraints of metric axioms, i.e. non-negativity, identity of indiscernibles, symmetry and triangle inequality. Some non-metric similarity works such as Tan et al. (2006) suggest that these restrictions are not compatible with human perception. As an example, they showed that although a centaur might be visually similar to both a person and a horse, the person and the horse are not similar to each other. A possible explanation for this phenomenon is that when comparing two images, human beings may pay more attention to similarities and thus, similar portions of the images may be more discriminative than dissimilar parts. To overcome the issues associated with applying strong rigid constraints to visual similarity, we propose to learn the non-metric similarity function $g$ using a neural network approach.

### 3.2 Image Representation

Here we describe the image representation method, $f$, we use. As this work aims to learn a non-metric similarity estimation from visual data, our efforts are not focused on improving existing image representation methods, but to learn how to compare them. Without loss of generality, we use

Table 1: Network architectures. Fully connected layers (FC-{filters}) are always followed by a ReLU layer except for the last one. Training: 22.5 million pairs. Validation: 7.5 million pairs.

|   | Config | Params | Training Data | | Validation Data | |
|---|--------|--------|-----|-----|-----|-----|
|   |        |        | MSE | $\rho$ | MSE | $\rho$ |
| A | FC-1024, FC-1024, FC-1 | 2.1M | 0.00021 | 0.946 | 0.00035 | 0.909 |
| B | FC-4096, FC-4096, FC-1 | 21M | 0.00008 | 0.978 | 0.00019 | 0.965 |
| **C** | FC-8192, FC-8192, FC-1 | 76M | **0.00007** | **0.982** | **0.00012** | **0.974** |
| D | FC-4096, FC-4096, FC-4096, FC-1 | 38M | 0.00009 | 0.978 | 0.00019 | 0.964 |

the RMAC descriptor proposed in Tolias et al. (2016) as image representation, although any other image representation method can be considered as well.

RMAC is a deep global image representation obtained from the last convolutional layer of a pre-trained CNN on ImageNet classification task Russakovsky et al. (2015). When an image is fed into the network, the last convolutional layer outputs a $W \times H \times K$ response, where $K$ is the number of filters and $W$ and $H$ are the spatial width and height of the output, respectively, that depend on the network architecture as well as on the size of the input image. The response of the $k$-th filter of the last convolutional layer can be represented by $\Omega_k$, a 2D tensor of size $W \times H$. If $\Omega_k(p)$ is the response at a particular position $p$, and $R$ is a spatial region within the feature map, the regional feature vector $f_R$ is defined as:

$$f_R = [f_{R,1} \ldots f_{R,k} \ldots f_{R,K}]^\top \tag{2}$$

where $f_{R,k} = \max_{p \in R} \Omega_k(p)$. Thus, $f_R$ consists of the maximum activation of each filter inside the region $R$. Several regional features are extracted at different multi-scale overlapping regions. Each of these regional vectors is independently post-processed with $\ell 2$-normalization, PCA-whitening and $\ell 2$-normalization, as suggested in Jégou & Chum (2012). Finally, regional vectors are summed and $\ell 2$-normalized once again to obtain the final compact vector. The size of the final vector is $K$, which is independent of the size of the input image, its aspect ratio or the number of regions used.

### 3.3 SIMILARITY NETWORK

To compare two images and obtain a visual similarity score we learn the similarity function $g$ by training a deep learning architecture. Given two input images $I_i$ and $I_j$, we first extract their representations $x_i$ and $x_j$, respectively, as explained in Section 3.2. The two $K$-dimensional global vectors are concatenated and fed into the similarity network, as shown in Figure 1. This process is different to the standard siamese architecture Chopra et al. (2005) because the latter maps images into vector representations and updates the shared weights according to the learning protocol and our approach trains and updates the similarity network on top of high-quality vector representations. Moreover, in the similarity network architecture, weights in the image representation block are not necessarily shared. Figure 2 shows the difference between both approaches.

The similarity network is composed by a set of fully connected layers, each one of them followed by a non-linear function, such as ReLU Krizhevsky et al. (2012). The input of the network is fixed to be of $1 \times K \times 2$ size, so the size of the first layer is $1 \times K \times 2 \times Ch$, where $Ch$ is the number of channels. We consider hidden layers of size $1 \times Ch \times 2 \times Ch$. Finally, the output layer is of size $1 \times Ch \times 2 \times 1$ and it is not followed by a ReLU layer, as the output similarity score is expected to cover a full range of values, both positive and negative. The regression loss function, $L$, penalizes when the predicted score of the network is far away from an annotated similarity score, such as:

$$L(I_i, I_j) = |s_{i,j} - y_{i,j}| = |g(x_i, x_j, w_g) - y_{i,j}| \tag{3}$$

where $s_{i,j}$ is the network output and $y_{i,j}$ is the annotated score. Four configurations A-D with different number of filters $Ch$ and number of hidden layers are proposed and tested during our experiments, as shown in Table 1.

### 3.4 TRAINING SIMILARITY

The visual similarity network is trained in three stages. In each stage the weights are initialized by the trained weights of the previous stage while the learned task gets progressively more difficult.

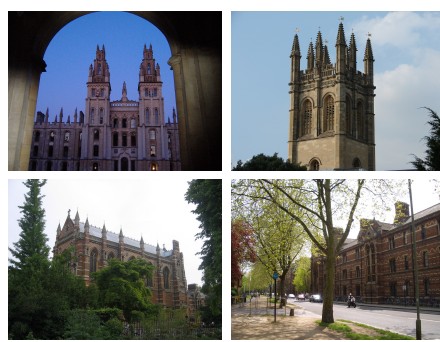 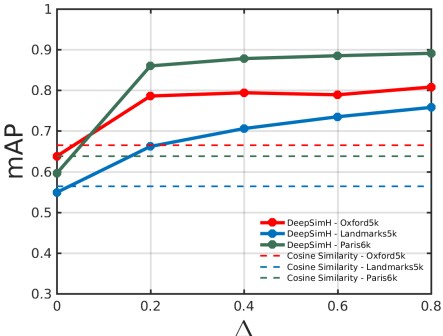

Figure 3: Misclassified pairs. (Upper) Lower row: (dissimilar) similar images in which the network score is (lower) higher than the cosine similarity.

Figure 4: mAP versus $\Delta$. Rigid lines are Deep-SimH scores, dashed lines are cosine similarity scores.

STAGE 1: STANDARD METRIC

In Stage 1, the network learns a standard similarity function based on the cosine similarity. We generate random pairs of vectors, $x_i$ and $x_j$, and we assign the cosine similarity between them as the score label $y_{i,j}$:

$$y_{i,j} = \frac{x_i \cdot x_j}{\|x_i\|\|x_j\|} \tag{4}$$

In order to train the model in the full range of possible values, pairs are produced so that the cosine similarity is uniformly distributed within the training set.

STAGE 2: VISUAL SIMILARITY

In Stage 2, the basic similarity network learns to increase the similarity score when given two matching images and to decrease it when a pair of images is not a match. The weights in this training stage are initialized by the weights obtained during Stage 1. We now use pairs of image representation vectors $x_i$ and $x_j$, randomly chosen from our training image dataset. The score label is set to:

$$y_{i,j} = \begin{cases} \frac{x_i \cdot x_j}{\|x_i\|\|x_j\|} + \Delta, & \text{if } x_i \text{ and } x_j \text{ are similar} \\ \frac{x_i \cdot x_j}{\|x_i\|\|x_j\|} - \Delta, & \text{otherwise} \end{cases} \tag{5}$$

where $\Delta$ is the margin parameter. Thus, the model learns to discriminate when a pair of images are similar (dissimilar) and assigns it a higher (lower) value than the standard score.

In this stage, the model learns how to compute a similarity score from examples of images that are known to be matching or non-matching. Therefore a relevant dataset to the final retrieval task should be used. Similarity between pairs might be decided using different techniques, such as image classes, score based on local features or manual labeling, among others. Without loss of generality, we consider two images as similar when they belong to the same class and as dissimilar when they belong to different classes.

STAGE 3: HARD EXAMPLES

In the Stage 3, the similarity network is refined by training it specifically by using *difficult pairs* of images. Previous works Gordo et al. (2016); Radenović et al. (2016) have shown that fine-tunning neural networks using difficult samples is very helpful in terms of performance. This is easy to understand: if the network is only trained by using *easy pairs* (e.g. a car and a dog), it will not be able to discriminate between *difficult pairs* (e.g. a car and a van). To choose the set of hard pairs we compute the scores of a random set of image pairs by using the network trained in Stage 2. Those pairs in which the network output is worse than the cosine similarity measure are selected as difficult pairs for retraining[1]. Examples of difficult image pairs can be seen in Figure 3.

---

[1] A worse score is a score that is lower in the case of a match and higher in the case of a non-match.

## 4 EXPERIMENTS

### 4.1 TESTING DATASETS

Our approach is evaluated on the standard image retrieval datasets described below.

**Oxford5k** Philbin et al. (2007): a dataset that consists of 5,062 images of 11 different Oxford landmarks. The query set contains 55 annotated images, 5 per landmark.

**Paris6k** Philbin et al. (2008): a datasets that consists of 6,412 images of 11 different Paris landmarks. The query set contains 55 annotated images, 5 per landmark.

**Land5k**: a validation subset of the Landmarks database Babenko et al. (2014). It consists of the 4,915 validation images from 529 classes. A random selection of 45 images is used as queries.

**Oxford105k**, **Paris106k**: the large-scale versions of Oxford5k and Paris6k, respectively. They include 100,000 distractor images from Flickr Philbin et al. (2007).

In both the Oxford5k and the Paris6k collections query images are cropped according to the region of interest provided by the authors of the datasets. Evaluation is performed by computing the mean Average Precision (mAP), using the provided ground truth and algorithms. For Land5k we consider an image to be relevant to the query when it belongs to the same class.

### 4.2 TRAINING DATASETS

For the purposes of this work, having a training dataset as similar as possible to the final similarity task is essential. We create several versions of the training dataset to evaluate the effect of using different samples in the training process.

**Landmarks** Gordo et al. (2016): an automatically cleaned subset of the full Landmarks Babenko et al. (2014) dataset which officially contains about 49,000 images from 586 landmarks. However, due to broken URLs, we could only download 33,119 training images and 4,915 validation images. This dataset does not contain images from classes that overlap with Oxford5k and Paris6k datasets as they were manually removed.

**Landmarks-extra500**: the Landmarks collection plus 250 random images from each of the Oxford5k and Paris6k datasets. In total, it contains 33,619 training images.

**Landmarks-extra**: the Landmarks collection in addition to about 500 images from Oxford5k and 1,700 images from Paris6k classes. In total, it contains 35,342 training images belonging to 605 different landmarks. Note that query images are not added in any case and they remain unseen by the system.

### 4.3 EXPERIMENTAL DETAILS

**Image Representation.** To compute RMAC representations we use the VGG16 network Simonyan & Zisserman (2014), which has been previously pre-trained on the ImageNet dataset Russakovsky et al. (2015). Unless otherwise stated, we use the default values proposed in Tolias et al. (2016) to obtain 512-dimensional RMAC vectors. VGG16 network is used off-the-shelf without any retraining or fine-tunning performed on top of it. Experimental results have shown that RMAC representations are very sensitive to the PCA matrices used in the post-processing step. As we are keeping query images unseen by the system and not using them in the PCA matrices computation as in Tolias et al. (2016), our results are slightly different to theirs.

**Visual Similarity Learning.** Similarity learning is trained using almost a million of random pairs, of which half of the pairs are visual matches and the other half are non-matches. PCA whitening is done using Paris5k images. As RMAC representation performs better in high resolution images, we re-scale all the images up to 1024 pixels, keeping the original aspect ratio of the pictures. For the similarity network, four different configurations A-D (Table 1) are explored during our experiments. The network is optimized using backpropagation and stochastic gradient descent. We use a learning rate of 0.001, a batch size of 100, a weight decay of 0.0005 and momentum of 0.9.

Table 2: mAP when using different training configurations and $\Delta$ (in brackets) values.

| | Landmarks | | | Landmarks-extra500 | | | Landmarks-extra | | |
| --- | --- | --- | --- | --- | --- | --- | --- | --- | --- |
| | Ox5k | Pa6k | La5k | Ox5k | Pa6k | La5k | Ox5k | Pa6k | La5k |
| Cosine | **0.665** | 0.638 | 0.564 | 0.665 | 0.638 | 0.564 | 0.665 | 0.638 | 0.564 |
| DeepCosine | 0.638 | 0.596 | 0.549 | 0.638 | 0.596 | 0.549 | 0.638 | 0.596 | 0.549 |
| OASIS | 0.514 | 0.385 | 0.578 | 0.570 | 0.651 | 0.589 | 0.619 | 0.853 | 0.579 |
| Linear (0.2) | 0.598 | **0.660** | 0.508 | 0.611 | 0.632 | 0.514 | 0.602 | 0.581 | 0.502 |
| DeepSim (0.2) | 0.658 | 0.460 | 0.669 | 0.717 | 0.654 | 0.671 | 0.718 | 0.757 | 0.668 |
| DeepSimH (0.2) | 0.655 | 0.503 | 0.697 | **0.719** | 0.677 | 0.693 | 0.786 | 0.860 | 0.662 |
| DeepSimH (0.4) | 0.637 | 0.504 | 0.737 | 0.703 | 0.701 | 0.745 | 0.794 | 0.878 | 0.706 |
| DeepSimH (0.6) | 0.613 | 0.514 | 0.776 | 0.703 | **0.716** | 0.776 | 0.789 | 0.885 | 0.735 |
| DeepSimH (0.8) | 0.600 | 0.511 | **0.783** | 0.685 | 0.710 | **0.803** | **0.808** | **0.891** | **0.758** |

**Computational cost.** Standard metrics are relatively fast and computationally cheap. Our visual similarity network involves the use of millions of parameters that inevitable increase the computational cost. However, it is still feasible to compute in a reasonable amount of time. In our experiments, training time is about 5 hours in a GeForce GTX 1080 GPU and testing time for a pair of images is 1.25 ms on average (0.35 ms when using cosine similarity).

## 5 RESULTS

### 5.1 ARCHITECTURE DISCUSSION

Four different configurations A-D for the similarity neural network are proposed. We compare the performance of each one during Stage 1, when the network is trained with the standard cosine similarity measurement. If $s_l$ is the network score and $y_l$ is the cosine similarity of the $l$-th pair with $l = 1..L$, we evaluate each network by computing the mean squared error, MSE, and the correlation coefficient, $\rho$, as:

$$MSE = \frac{1}{L} \sum_{l=1}^{L} (s_l - y_l)^2 \qquad\qquad \rho = \frac{1}{L-1} \sum_{l=1}^{L} \frac{s_l - \mu_s}{\sigma_s} \frac{y_l - \mu_y}{\sigma_y} \qquad (6)$$

where $\mu_s$ and $\sigma_s$ are the mean and standard deviation of the vector of network scores $s$, and $\mu_y$ and $\sigma_y$ are the mean and standard deviation of the vectors of cosine similarities $y$.

Results are shown in Table 1. Unsurprisingly, the configuration with bigger number of parameters, C, achieves the best MSE and $\rho$ results, both in training and validation sets. However, the performance of networks B and D is very close to the performance of network C. As network B requires only 21 million parameters and network C requires 76 million parameters, we keep configuration B as our default architecture for the rest of the experiments.

### 5.2 EVALUATION OF THE SIMILARITY NETWORK

In this section, we study the benefits of using a non-metric distance function trained with neural networks. In order to isolate the contribution of the visual similarity computation and perform a fair comparison between different distance functions, we only train the similarity network part. However, an end-to-end training of the whole image retrieval pipeline is explored in Appendix B.

To evaluate our similarity network, we compute the mAP at each stage of the training process (Section 4.2). Results when using different training datasets can be found in Table 2. Cosine similarity is computed as a baseline. We denote as DeepCosine the results obtained after the first stage, when the network is trained to mimic cosine similarity. Naturally, DeepCosine performs worse than the cosine similarity, as it is an estimation of the cosine metric. DeepSim refers to the results obtained after the second stage, when the network is fine-tunned to learn visual similarity with random pairs of images. DeepSimH are the results after the last stage, when the network is trained by using both random and hard pairs of images. We compare our approach against the standard similarity learning algorithm OASIS Chechik et al. (2010). Finally, we also conduct experiments on linear metric learning, which are denoted as Linear in Table 2, by training an affine transformation of the feature vectors using the same training protocol as described in Equation 5.

Table 3: mAP results for different state-of-the-art methods. Dim corresponds to the dimensionality of the feature representation. Similarity is the similarity function.

| | Method | Dim | Similarity | Ox5k | Ox105k | Pa6k | Pa106k |
|---|---|---|---|---|---|---|---|
| Off-the-shelf | Babenko et al. (2014) | 512 | L2 | 0.435 | 0.392 | - | - |
| | Sharif Razavian et al. (2014) | 4096 | Averaged L2 | 0.322 | - | 0.495 | - |
| | Wan et al. (2014) | 4096 | OASIS | 0.466 | - | 0.867 | - |
| | Babenko & Lempitsky (2015) | 256 | Cosine | 0.657 | 0.642 | - | - |
| | Yue-Hei Ng et al. (2015) | 128 | L2 | 0.593 | - | 0.59 | - |
| | Kalantidis et al. (2016) | 512 | L2 | 0.708 | 0.653 | 0.797 | 0.722 |
| | Mohedano et al. (2016) | 25k | Cosine | 0.739 | 0.593 | 0.82 | 0.648 |
| | Salvador et al. (2016) | 512 | Cosine | 0.588 | - | 0.656 | |
| | Tolias et al. (2016) | 512 | Cosine | 0.669 | 0.616 | 0.83 | 0.757 |
| | Jiménez et al. (2017) | 512 | Cosine | 0.712 | 0.672 | 0.805 | 0.733 |
| | Ours ($\Delta = 0.8$) | 512 | DeepSimH | **0.808** | **0.772** | **0.891** | **0.818** |
| Fine-tunning | Babenko et al. (2014) | 512 | L2 | 0.557 | 0.522 | - | - |
| | Gordo et al. (2016) | 512 | Cosine | 0.831 | 0.786 | 0.871 | 0.797 |
| | Wan et al. (2014) | 4096 | OASIS | 0.783 | - | **0.947** | - |
| | Radenović et al. (2016) | 512 | Cosine | 0.77 | 0.692 | 0.838 | 0.764 |
| | Salvador et al. (2016) | 512 | Cosine | 0.71 | - | 0.798 | - |
| | Gordo et al. (2017) | 2048 | Cosine | 0.861 | **0.828** | 0.945 | **0.906** |
| | Ours ($\Delta = 0.8$) | 512 | DeepSimH | **0.882** | 0.821 | 0.882 | 0.829 |

Our similarity networks outperform OASIS in all the testing datasets. Moreover when using Landmarks-clean-extra as training dataset, results are boosted with respect to the standard metric, achieving improvements ranging from 20% (Oxford5k) to 40% (Pairs6k). When using a small subset of images from Oxford5k and Paris6k classes, i.e. Landmarks-clean-extra-500 dataset, our similarity networks also improve mAP with respect to the cosine similarity in the three testing datasets. This indicates that visual similarity can be learnt even when using a reduced subset of the target image domain. Experiments on affine transformations show that, unlike our proposed methods, simple linear metrics are not able to properly fit Equation 5. However, visual similarity does not transfer well across domains when no images of the target domain are used during training. An extended discussion about the effects of the training dataset can be found in Appendix A.

Overall, these results suggests that our network is able to learn whether two images are similar or not and provide a similarity score accordingly. Figure 4 shows how the mAP is affected when using different values of $\Delta$. Except when $\Delta = 0$ (i.e. visual similarity is not learned), DeepSimH always improves mAP with respect to the standard cosine similarity.

## 5.3 COMPARISON WITH THE STATE OF THE ART.

Finally, we compare our method against several state-of-the-art techniques (Table 3). As standard practice, works are split into two main groups: off-the-shelf and fine-tunning approaches. Off-the-shelf are techniques that extract visual representations by using CNNs trained on ImageNet dataset Russakovsky et al. (2015) without modifying the network. On the other hand, fine-tunning methods retrain the network to compute more accurate visual representation. For a fair comparison, we only consider methods that represent each image with a single compact vector and do not apply query expansion or image re-ranking. When using off-the-shelf RMAC features, our DeepSimH approach outperforms previous methods in every dataset. To compare against fine-tunned methods, we compute RMAC vectors using the fine-tunned version of VGG16 proposed in Radenović et al. (2016) and training our DeepSimH exactly in the same way as in the off-the-shelf version. Accuracy is significantly improved when using our similarity network instead of the analogous cosine similarity method Radenović et al. (2016). DeepSimH achieves the best mAP precision in Ox5k dataset and comes second in Ox105k and Pa106k after Gordo et al. (2017), which uses the more complex and higher-dimensional ResNet He et al. (2016) instead of a VGG16 network for image representation.

## 6 CONCLUSIONS

We have presented a method for learning visual similarity directly from visual data. Instead of using a rigid metric distance, such as the standard cosine similarity, we propose to train a neural network

model to learn a similarity estimation between a pair of visual representations previously extracted from input images. Our method outperforms state-of-the-art approaches based on rigid distances in standard image retrieval collection of images and experimental results showed that learning a non-metric visual similarity function is beneficial in image retrieval tasks provided that a small subset of images of the same domain are available during training. Standard image retrieval techniques that are commonly applied after cosine similarity computation, such as query expansion or image re-ranking, might also be applied on top of the similarity network. Finally, we end with an open question, which is the subject of planned future work, concerning efficient computation of exact or approximate K-nearest neighbours based on the learned network similarity function.

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

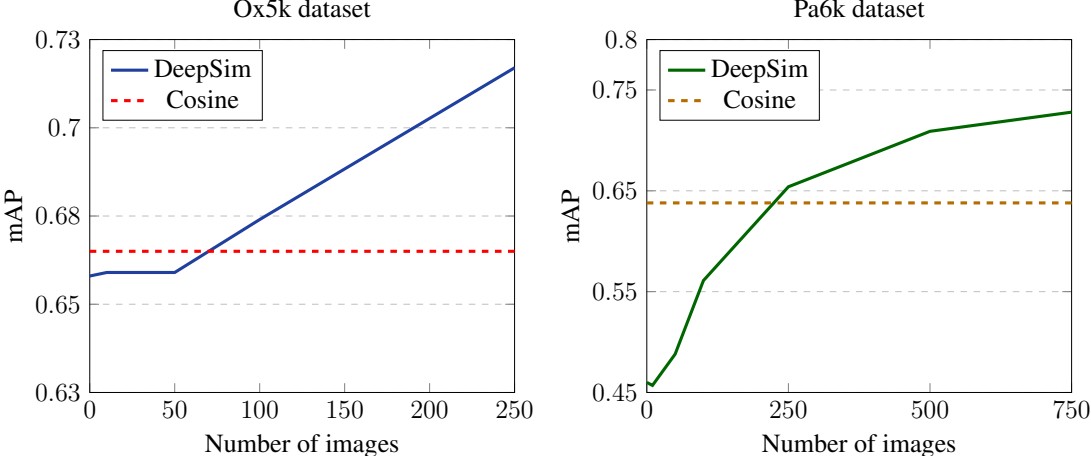

Figure 5: mAP when using different number of target samples in the training set.

## APPENDIX A    TRAINING ON TARGET DATASET

In this appendix, a further discussion about the influence of the dataset used to train the similarity network and estimate the visual similarity between a pair of images is carried out.

As we already noted in Section 5.2, visual similarity does not transfer well across domains. A subset of samples from the target dataset is required during training to learn a meaningful similarity function. This is a well-known problem in the field of metric learning (Kulis et al. (2013)). In Figure 5, we explore the effect on performance when we use different subsets of samples from the target collection in addition to the Landmarks dataset (Gordo et al. (2016)) during the second stage of our training (Section 3.4).

Figure 5 shows that there is a clear correlation between the similarity network performance and the number of samples from the target dataset used during training. Indeed, in agreement with previous work in metric learning (Kulis et al. (2013)), we observe that not considering samples from the target dataset to train a similarity function might be harmful. The similarity network, however, outperforms standard metric results even when a small number of samples from the target collection is used during training: only 100 images from Ox5k and 250 images from Pa6k are required to outperform cosine similarity in Ox5k and Pa6k datasets, respectively. This fact suggests that the similarity network is able to generalize from a small subset of target samples and is not memorizing the distances in the training collection.

Finally, we present some visual results of our findings. Figure 6 and Figure 7 show the t-Distributed Stochastic Neighbor Embedding (t-SNE) (Van Der Maaten, 2014) representation of Ox5k images when using RMAC as image representation, and cosine similarity or our similarity network as similarity function, respectively. Although RMAC descriptor with a standard metric is already performing well in terms of visual similarity (e.g., in Figure 6 images from Radcliffe camera are grouped together in the right bottom corner), performance can be pushed even more when our similarity network is used instead (Figure 7. In summary, these results indicate the benefit of training a similarity network over a standard metric function such as cosine similarity for the image retrieval task.

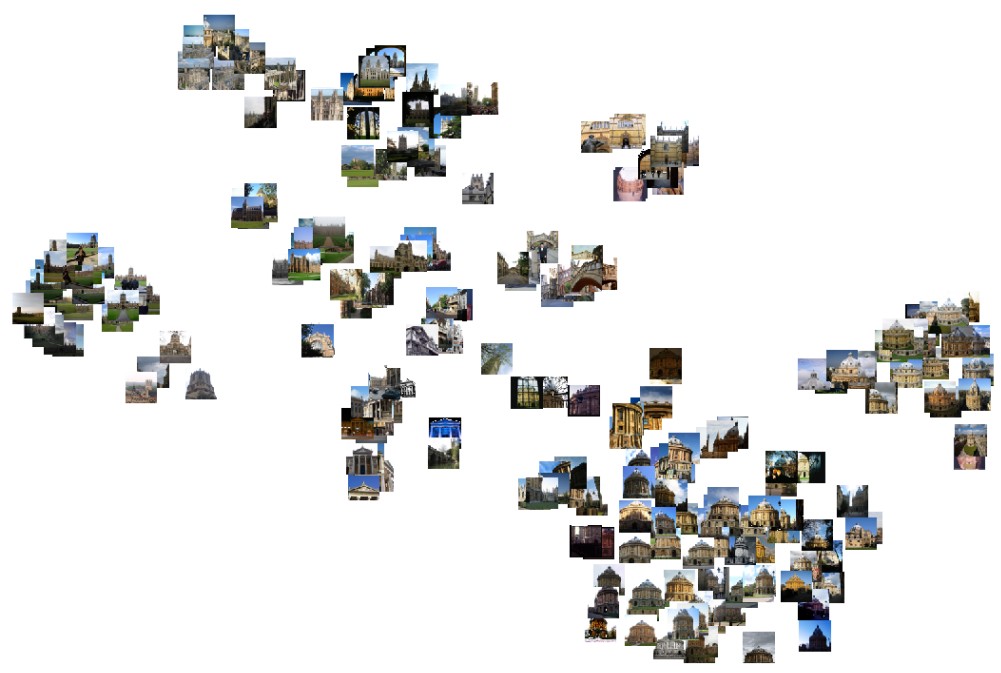

Figure 6: t-SNE plot for a subset of 500 Ox5k images when using RMAC and cosine similarity.

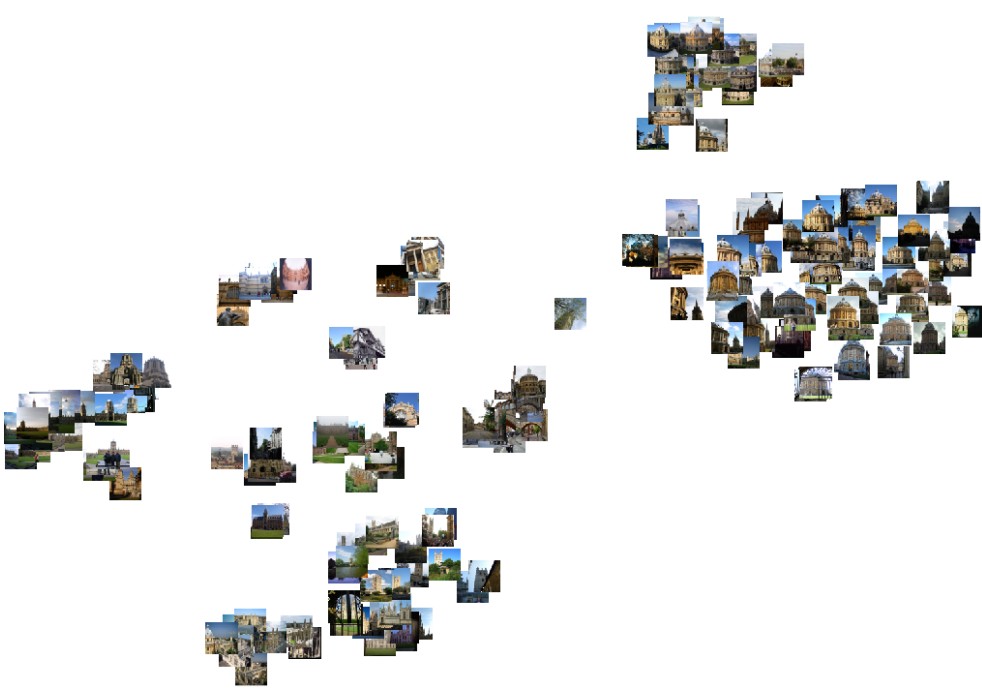

Figure 7: t-SNE plot for a subset of 500 Ox5k images when using RMAC and DeepSim.

## APPENDIX B END-TO-END TRAINING

So far, we have isolated the similarity computation part in the image retrieval pipeline by only training the similarity network. In this way, it is easy to see that the improvement in the testing datasets compare to when using other similarity methods (Section 5.2) is, in fact, due to the visual similarity network function. In this appendix, however, we explore a real end-to-end approach for image retrieval. The end-to-end approach consists on feeding the system with pixels to obtain a visual similarity score between a pair of images. The whole pipeline is presented in Figure 8. For the feature extraction part, we adopt the MAC compact image representation, following Radenović et al. (2016) work. For the visual similarity part, we use our visual similarity network DeepSim. The whole approach is end-to-end differentiable so backpropagation can be applied during training.

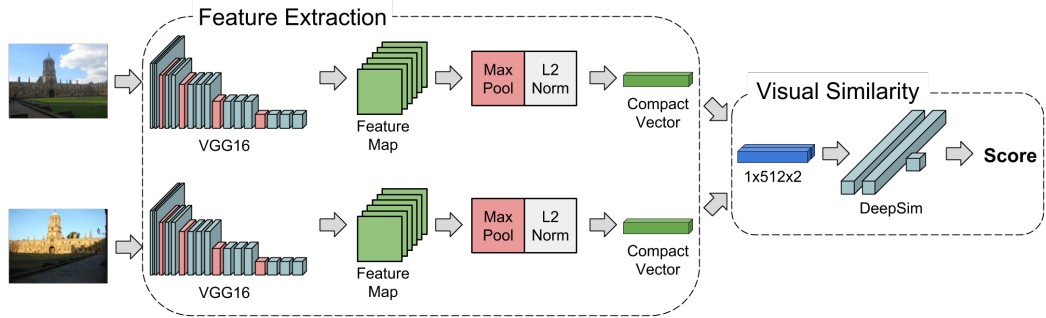

Figure 8: End-to-End architecture. The feature extraction part consists on a VGG16 network followed by a max-pooling and a l2-normalization layers. In the visual similarity part, two compact vectors are concatenated and forwarded to the DeepSim network to obtain a similarity score.

In this case, we use MAC Tolias et al. (2016) as compact image representation. After feeding a VGG16 network Simonyan & Zisserman (2014) with a pre-processed image, the feature maps from the last convolutional layer are obtained. These feature maps are then max-pooled over the whole region to obtain a compact vector, which is l2-normalized. The final dimensionality of the MAC vector does not depend on the input image size, but in the number of filters in the last convolutional layer. Image pre-processing includes resizing the image to 720 pixels on its largest side (maintaining aspect ratio) and mean subtraction.

We initialize the VGG16 network with the weights trained on ImageNet dataset. We then learn the weights of the similarity network by freezing VGG16 weights and applying Stage 1 and Stage 2, as described in Section 3.4. Finally, for the end-to-end training, we unfreeze all the weights of the architecture and fine-tune all the layers one last time. As all the layers have been already pre-trained, the final end-to-end fine-tunning is performed in about 200,000 pairs of images from Landarmarks-extra dataset (Section 4.2) for just 5,000 iterations. Note that we adopt MAC Tolias et al. (2016) instead of RMAC as it is easier to train and thus, the results are slightly worst. From Table 4 we note, firstly, a boost in performance when using DeepSim instead of the cosine similarity and finally, a significant improvement when the architecture is trained end-to-end with respect to both the baseline and when only training the visual similarity part.

The results are unsurprising as fine-tuning the entire architecture allows us to fit better to a particular dataset. However the key message of the paper is that fine-tuning the final similarity computation, instead on relying on cosines as researchers have been doing so far, may be a worthwhile step that can push accuracy results higher irrespective of the feature vector computation.

Table 4: mAP when training different parts of the image retrieval pipeline. In blue, the modules that are fine-tunned in every experiment.

| Features | Similarity | Oxford5k | Paris6k | Landmarks5k |
|----------|-----------|----------|---------|-------------|
| MAC | Cosine | 0.481 | 0.539 | 0.494 |
| MAC | **DeepSim** | 0.509 | 0.683 | 0.589 |
| **MAC** | **DeepSim** | 0.555 | 0.710 | 0.685 |

