# OpenReview forum: "Learning Non-Metric Visual Similarity for Image Retrieval"
_ICLR.cc/2018/Conference — Reject_

### Official Review · AnonReviewer2 · 2017-11-27
**Pushing the performance in image retrieval by learning a non-metric similarity**

**Rating:** 7
**Confidence:** 5

**Review:**

The authors of this work propose learning a similarity measure for visual similarity and obtain, by doing that, an improvement in the very well-known datasets of Oxford and Paris for image retrieval. The work takes high-level image representations generated with an existing architecture (R-MAC), and train on top a neural network of two fully connected layers.

The training of such network is performed in three stages: firstly approximating the cosine similarity with a large amount of random feature vectors, secondly using image pairs from the same class, and finally using the hard examples.


PROS

P1. Results indicate the benefit of this approach in terms of similarity estimation and, overall, the paper present results that extend the state of the art in well-known datasets.

P2. The authors make a very nice effort in motivation the paper, relating it with the state of the art and funding their proposal on studies regarding human visual perception. The whole text is very well written and clear to follow.

CONS

C1. As already observed by the authors, training a similarity function without considering images from the target dataset is actually harmful. In this sense, the simple cosine similarity does not present this drawback in terms of lack of generalization. This observation is not new, but relevant in the field of image retrieval, where in many applications the object of interest for a query is actually not present in the training dataset.

C2. The main drawback of this approach is in terms of computation. Feed-forwarding the two samples through the trained neural network is far more expensive that computing the simple cosine similarity, which is computed very quickly with a GPU as a matrix multiplication. The authors already point at this in Section 4.3.

C3. I am somehow surprised that the authors did not explore also training the network that would extract the high-level representations, that is, a complete end-to-end approach. While I would expect to have the weights frozen in the first phase of training to miimic the cosine similarity, why not freeing the rest of layers when dealing with pairs of images ?

C4. There are a couple of recent papers that include results of the state of the art which are closer and sometimes better than the ones presented in this work. I do not think they reduce at all the contribution of this work, but they should be cited and maybe included in the tables:

A. Gordo, J. Almazan, J. Revaud, and D. Larlus. End-to-end learning of deep visual representations for image retrieval.
International Journal of Computer Vision, 124(2):237–254, 2017.

Albert Jimenez, Jose M. Alvarez, and Xavier Giro-i-Nieto. “Class-Weighted Convolutional Features for Visual Instance Search.” In Proceedings of the 28th British Machine Vision Conference (BMVC). 2017.

---

> ### Author Response · Authors · 2017-12-06
> **Authors' response to Reviewer 2**
>
> Thank you very much for your useful suggestions. We'd like to address your comments for further improvement of our work.
>
> C1. We agree and we actually observe this phenomenon in our experiments. However, we found that our similarity network generalizes well even when few samples of the target dataset are given (for example, in the Oxford dataset, with only 100 samples from the target dataset our similarity network outperforms cosine similarity). Further details on this topic are going to be included as an appendix in the next update of the paper.
>
> C2. As already stated in the paper, standard metrics are relatively fast and computationally cheap. However, we believe that computation might not be necessarily a problem with the current computational power of GPUs. Moreover, speeding up the network similarity computation and linking it to some approximate nearest neighbour shceme is one of the things we are currently looking at.
>
> C3. We also believe that a end-to-end training is the next step after the results of this work. It is for sure a very interesting experiment to perform. However, the scope of this paper was to study the benefits of using a non-metric distance function trained with neural networks. In order to isolate the contribution of the similarity computation part and perform a fair comparison between distance functions, we used standardized features (R-MAC) as image descriptors. By training the whole pipeline in an end-to-end way we would have never found which part of the improvement was because of the feature extraction fine-tunning and which part of the improvement was due to the non-metric similarity computation.
>
> C4. We apologize for any missing citation. Thank you for pointing us to a couple of missing works, which for sure are going to be included in an updated version of the paper.

---

### Official Review · AnonReviewer3 · 2017-11-28
**Motivation is unclear and more evaluations are needed**

**Rating:** 4
**Confidence:** 4

**Review:**

(1) The motivation
The paper argues that it is more suitable to use non-metric distances instead of metric distances. However, the distance function used in this work is cosine similarity between two l2 normalized features. It is known that in such a situation, cosine similarity is equivalent to Euclidean distance. The motivation should be further explained.

(2) In Eq. (5), I am not sure why not directly set y_ij = 1 if two images come from the same category, and set to 0 otherwise. It is weird to see the annotation is related to the input features considering that we already have the groundtruth labels.

(3) The whole pipeline is not trained in an end-to-end manner. It requires some other features as the input (RMAC used in this work), and three-stage training. It is interesting to see some more experiments where image pixels are the input.

(4) The algorithm is not comparable to the state-of-the-art. Some representative papers have reported much better performances on the datasets used in this paper. It is suggested to refer to some recent papers in top conferences.

---

> ### Author Response · Authors · 2017-12-06
> **Authors' response to Reviewer 3**
>
> Thank you for the review. We apologize if the first draft was unclear in certain aspects. Below, we'd like to clarify and address all of your points.
>
> (1) The motivation of our work can be summarized as in the following paragraph of the paper (page 3):
>
> "Note that g does not have to be a metric in order to be a similarity function and thus, it is not required to satisfy the rigid constraints of metric axioms, i.e. non-negativity, identity of indiscernibles, symmetry and triangle inequality. Some non-metric similarity works such as Tan et al. (2006) suggest that these restrictions are not compatible with human perception. As an example, they showed that although a centaur might be visually similar to both a person and a horse, the person and the horse are not similar to each other. A possible explanation for this phenomenon is that when comparing two images, human beings may pay more attention to similarities and thus, similar portions of the images may be more discriminative than dissimilar parts. To overcome the issues associated with applying strong rigid constraints to visual similarity, we propose to learn the non-metric similarity function g using a neural network approach."
>
> The basic idea behind these lines is that the human perception of visual similarity might not correspond to what a linear metric, such as cosine similarity or Euclidean distance, represents. Thus, our work proposes to learn a non-metric similarity function with a convolutional neural network. In our work, we do not use the cosine similarity between two l2 normalized features as you stated, but the non-metric similarity function trained in our model. Then, this non-metric similarity function is applied to any pair of visual features *instead* of a standard metric (such as cosine similarity) to rank images by score in image retrieval problems. We argue and show in our experimentation that by using our non-metric similarity function we can push performance in standard image retrieval datasets.
>
>
> (2) In contrast to classification methods where labels are discrete (in this case, 1 if images are similar or 0 otherwise), a similarity function for image retrieval ranking should produce a continuous set of scores. This is for obvious reasons: even within the same class, a pair of images might be more similar than another pair of images, and the similarity function should reflect this behavior in the output scores.
>
>
> (3) We also believe that a end-to-end training is the next step after the results of this work. It is for sure a very interesting experiment to perform. However, the scope of this paper was to study the benefits of using a non-metric distance function trained with neural networks. In order to isolate the contribution of the similarity computation part and perform a fair comparison between distance functions, we used standardized features (R-MAC) as image descriptors. By training the whole pipeline in an end-to-end way we would have never found which part of the improvement was because of the feature extraction fine-tunning and which part of the improvement was due to the non-metric similarity computation.
>
>
> (4) We apologize for any missing citation and we are willing to include any missed work in an updated version of the paper. So far, we are aware of the following papers [1], [2] and [3]. However, we do not consider that these papers reduce the contribution of our work, as the assumption that using a similarity network is beneficil in image retrieval is still validated through our extensive evaluation.
>
> [1] A. Gordo, J. Almazan, J. Revaud, and D. Larlus. End-to-end learning of deep visual representations for image retrieval. IJCV 2017.
> [2] A. Jimenez, J. M. Alvarez, and X. Giro-i-Nieto. “Class-Weighted Convolutional Features for Visual Instance Search.” BMVC 2017.
> [3] H. Noh, A. Araujo, J. Sim, T. Weyand, and B. Han. Large-Scale Image Retrieval With Attentive Deep Local Features. ICCV 2017.

---

### Official Review · AnonReviewer1 · 2017-12-09
**Lack of technical contribution**

**Rating:** 3
**Confidence:** 5

**Review:**

This paper presents a simple image retrieval method. Paper claims it is a deep learning method, however it is not an end-to-end network. The main issue of the paper is lack of technical contributions.

Paper assumes that image retrieval task can be reformulated at a supervised similarity learning task. That is fine, however image retrieval is traditionally an unsupervised task.

Even after using supervised method and deep learning technique, still this method is not able to obtain better results than hand crafted methods. Why is that? See - paper from CVPR2012 -  Arandjelović, Relja, and Andrew Zisserman. "Three things everyone should know to improve object retrieval." Computer Vision and Pattern Recognition (CVPR), 2012 IEEE Conference on. IEEE, 2012.

Paper make use of external signal to obtain y_{i,j}. It is not clear to me how does this generalize to large datasets?

If features are L2 normalized, why you need to normalize the features again in equation 5?

In equation 5, why not simply use a max margin deep similarity metric learning method with slack variables to generalizability?

The performance of entire network really rely on the accuracy of y_{i,j} and it is not clear the obtained performance is simply due to this supervision.

Paper does not argue well why we need this supervision.

Technically, there is nothing new here.

---

> ### Author Response · Authors · 2017-12-19
> **Authors' response to Reviewer 1**
>
> Thank you for your review. We really appreciate criticism in our work in order to keep improving it. Below, we'd like to address the points you raise.
>
> As already clarified in the other author responses, the paper is certainly not an end-to-end network for a very specific reason. We wanted to study the benefits of using a non-metric distance function trained with neural networks. This is something researchers have not looked at, to the best of our knowledge. Our results indicate that by casting the similarity computation as a trainable network there are benefits to be gained, irrespective of what feature extraction you use. Here we have used RMAC as a basis and shown how there are extra percentage points of accuracy to be gained by training a similarity computation instead of just using cosines. We believe the narrative of the paper gets a bit confused by including an end-to-end training but since all three reviewers mentioned it, we have indeed included the experiment in appendix B. As expected, doing end-to-end training offers even further improvement to the accuracy results but the story of the paper remains the same: “Whatever your image feature vector, consider fine-tuning your cosine similarity computation"
>
> Concerning the rest of your points. The assumption that image retrieval can be reformulated as a supervised task is not new and it is broadly assumed in some of the state-of-the art methods, such as:
>
> F. Radenovic, G. Tolias and O. Chum. CNN image retrieval learns from BoW: Unsupervised fine-tuning with hard examples. ECCV 2016.
>
> A. Gordo, J. Almazan, J. Revaud, and D. Larlus. End-to-end learning of deep visual representations for image retrieval. IJCV 2017.
>
> We are really confused when you say that our method is not able to obtain better results than hand-crafted methods. As it can be seen in Table 3, our method outperforms almost all the methods based in compact representations in image retrieval literature. We would appreciate if you can please clarify which methods are you referring to. As for query expansion and image re-ranking, these are add-ons that can still be applied on top of our similarity network in the same way as they are applied in top of cosine similarity. Such methods are not competitors as they might be applied altogether to push image retrieval performance.
>
> With respect to equation (5) and the cosine similarity computation, features are normalized as it is the standard procedure. We envisaged our method to be used with any kind of features, whether previously L2 normalized or not.
>
> Finally, experiments in larger datasets (Oxford 105k and Paris 106k) are conducted in Table 3.

---

### Author Response · Authors · 2017-12-21
**Comment on the new revision**

We would like to thank the reviewers for their feedback. Considering all their suggestions, we have tried our best to improve our work and we have uploaded a new version of the paper.

We would like to emphasized that the main idea of our work is to use a non-metric similarity function based on neural networks instead of a standard metric for image retrieval (cosines). We argue that the visual human perception is not properly explained by linear metrics and thus, non-metric visual similarity may obtain better performance than the standard cosine similarity in image retrieval systems. We think this is something researchers have not looked at, to the best of our knowledge.

We propose a simple approach based on a three stage training to learn a non-metric visual similarity and we perform exhaustive experiments to evaluate its performance. Experiments show that using a visual similarity function based on neural networks instead of cosine similarity is actually beneficial and can improve results in standard image retrieval datasets considerably. The paper is not intended to be an end-to-end network as we wanted to study the benefits of using a different similarity function than standard cosine similarity. We thought that the results of the paper would not have been clear about the benefits of our method by training an end-to-end approach, but since all the three reviewers agree that it is an interesting experiment, we have indeed included it in this revision.

In brief, the new version of the paper includes the following improvements:

1. We trained a simple linear metric method using the same data and training protocol as our proposed method. Results, included in Table 2, show that a linear system based on affine transformations can not fit the visual similarity between images as well as a nonlinear MLP. This suggests that the benefits in our system are due to the non-metric nature of the architecture and not to the training configuration, as suggested by Reviewer 1.

2. After the concerns of Reviewer 2 about the training dataset, we performed further empirical evaluation in this issue, which is now included in appendix A. Results show that the similarity function is able to generalize well even when a small subset of the target domain is used.

3. As all the three reviewers agree that an end-to-end experiment is very interesting, we included an end-to-end approach in appendix B. Unsurprisingly, the results show that an end-to-end approach is even more beneficial, as fine-tuning the entire architecture allows us to fit better to a particular dataset. However, we would like to emphasize that the key message of the paper is that fine-tuning the final similarity computation, instead on relying on cosines as researchers have been doing so far, may be a worthwhile step that can push accuracy results higher, irrespective of the feature vector computation.

4. In relation to the missing citations, we included the following missing work in Section 2 (Related Work) and Section 5.3 (Comparison with the state of the art):
- A. Gordo, J. Almazan, J. Revaud, and D. Larlus. End-to-end learning of deep visual representations for image retrieval. IJCV 2017.
- A. Jimenez, J. M. Alvarez, and X. Giro-i-Nieto. “Class-Weighted Convolutional Features for Visual Instance Search.” BMVC 2017.
- H. Noh, A. Araujo, J. Sim, T. Weyand, and B. Han. Large-Scale Image Retrieval With Attentive Deep Local Features. ICCV 2017.

Thank you for reading our response and please, consider re-reading the new version of the paper and updating your reviews, if appropriate.

---

### Decision · Program_Chairs · 2018-01-29
**ICLR 2018 Conference Acceptance Decision**

**Decision:**

Reject

**Comment:**

Two reviewers recommended rejection, and the last reviewer votes for acceptance. The authors provided a rebuttal, including the end-to-end experiment (although the AC agrees with the authors that this experiment is not crucial to the paper). The AC read the paper and the reviews. While there are clearly interesting aspects of this work, it somewhat falls short in terms of the technical contribution. Perhaps a better writing would alleviate this issue: for example, explaining the visual features is somewhat a distraction from the main point, and could be put in the end. The 3 stage training is somewhat ad hoc (or less elegant). Since there are many excellent papers submitted to ICLR this year, this paper unfortunately did not make it above the bar.